# The Effect of Functionally Graded Materials on Temperature during Frictional Heating: Under Uniform Sliding

**DOI:** 10.3390/ma14154285

**Published:** 2021-07-31

**Authors:** Aleksander Yevtushenko, Katarzyna Topczewska, Przemysław Zamojski

**Affiliations:** Faculty of Mechanical Engineering, Bialystok University of Technology (BUT), 45C Wiejska Street, 15-351 Białystok, Poland; a.yevtushenko@pb.edu.pl (A.Y.); zamojski.przemyslaw@gmail.com (P.Z.)

**Keywords:** frictional heating, functionally gradient materials, temperature, composite, ceramic

## Abstract

The mathematical model of heating process for a friction system made of functionally graded materials (FGMs) was proposed. For this purpose, the boundary-value problem of heat conduction was formulated for two semi-spaces under uniform sliding taking into consideration heating due to friction. Assuming an exponential change in thermal conductivities of the materials, the exact, as well as asymptotic (for small values of time), solutions to this problem were obtained. A numerical analysis was performed for two elements made of ZrO_2_–Ti-6Al-4V and Al_3_O_2_–TiC composites. The influence of the gradient parameters of both materials on the evolution and spatial distributions of the temperature were investigated. The temperatures of the elements made of FGMs were compared with the temperatures found for the homogeneous ceramic materials.

## 1. Introduction

A new class of composite materials with non-homogeneous spatial distribution of properties has emerged in recent years in the field of materials science [1]. Such properties are intentionally obtained during manufacturing by grading the internal structure of a material. Depending on the fabrication process, they are designed as stepwise-graded or continuous-graded materials [2]. The typical representatives of stepwise-graded composites are the laminates. The defect of such materials is the discontinuity of stress on the interfaces between adjacent discrete layers [3]. Materials with a continuous change in properties, known as functionally graded materials (FGMs) are devoid of this drawback. Nowadays, FGMs are usually a mixture of two distinct materials with continuously varying volume fractions of the constituents that, in effect, possess smooth properties which change along a certain direction [4]. Functionally graded materials possess a number of advantages that make them attractive in potential applications [5]. For example, a significant reduction of thermal stress in a heated element has been be achieved by introducing a thermal conductivity gradient in the material [6]. The results of studies indicate that a controlled continuous change in material properties can lead to a significant improvement in resistance to contact deformation and damage [4,7]. Thus, functionally graded coatings have been proposed as an alternative to replace conventional homogeneous coatings of frictional elements [8,9]. It has been proven that FGM coatings subjected to thermal shocks may suffer less damage than conventional ceramic coatings [6]. 

Usually, functionally graded materials are made of ceramic-metal composites and have superior characteristics of both components, i.e., heat and corrosion resistance of the ceramic and mechanical strength of the metal, at the same time [10]. Therefore, FGMs are considered to be advanced materials resistant to wear and elevated temperature conditions, and therefore they have great potential for use in heavy loaded sliding systems. One such application is brake discs exposed to intensive heating due to friction. At the core of an FGM disc, the material is steel to maintain structural rigidity, which gradually changes along the thickness and approaches purely ceramic at the friction surfaces to resist the severe thermal loading [5,11]. This significantly improves the thermomechanical behavior of the brake system as a whole [12].

Investigations associated with the development of frictional heating models for FGMs to determine the distributions of temperature and thermal stresses in brake systems, have received a great deal of attention from many researchers. The most common investigations have simulated the temperature regime in FGM brakes using numerical methods, in particular, by means of the finite element method (FEM). The finite element analysis of axisymmetric thermoelastic contact problems for a functionally graded disc with material property changes in the radial direction was performed by Shahzamanian et al. [13,14]. In [5], the corresponding problem was analyzed for a disc with properties dependent on the depth, along a normal direction to the friction surface of the disc. It was established that with the same operating parameters, the temperature gradient in a functionally graded disc was significantly lower than in a conventional steel disc. In a study by [9], the finite element methodology was used to compute the subsurface stresses in functionally graded coatings subjected to frictional contact with heat generation. 

In addition to the well-established finite element method, there are other numerical methods for solving the corresponding heat problems of friction for functionally gradient materials. An advanced computational method for transient heat conduction analysis in a non-homogeneous FGM, based on local boundary integral equations, was proposed by Sladek et al. [15]. The Green’s functions for the three-dimensional FGM transient heat conduction equation was derived using an exponential variation transform by [16]. The boundary integral equation based upon this approach has been solved numerically using a Galerkin approximation. The hybrid numerical method, based on the weighed residual and Fourier transform methods, to investigate the temperature distribution in the FGM plates under the exponential heat source load, was adopted by Tian and Jiang [17].

However, the closed-form analytical solutions to the thermal problems of friction for FGMs have higher accuracy and require less computational time than other methods. In general, the problems of thermomechanical contact with frictional heating for material with non-homogeneous properties are difficult to solve analytically due to the high mathematical complexity. For such materials, the equations of thermal conductivity and thermoelasticity contain coefficients that depend on the spatial coordinate [18]. Thus, the exact solutions of these equations and the determination of temperature distributions on their basis, require some special assumptions [19]. It is known that the superb performance of a functionally graded brake disc is achieved by introducing the appropriate gradient of thermomechanical properties by adjusting the gradient index [4]. The distribution of material properties in the FGM models is usually limited to unidirectional changes in the constituents of the composite [5]. There are two main distinctive ways to approximate the distribution of material properties through the graded direction, i.e., by means of an exponential and a power function. Note that the actual variations of properties depend on the material manufacturing process, which is neither exponential nor power law, therefore, in both cases, some level of curve fitting is implied [20]. However, both of these functions have a parameter that can be regulated to improve the fit and to adjust the gradation of the material. This role is played by the exponential decay rate in the exponential and the power exponent in the power law. The selection of these functions is also crucial from the point of view of the difficulty solving the thermal problems for FGMs by analytical methods.

The one-dimensional transient heat conduction problem for an axisymmetric FGM cylindrical shell with nonlinear thermal conductivity distributed according to the power law has been solved by the methods of separation of variables and Bessel functions [21]. The analytical formulas for calculating the thermal and mechanical stresses in a hollow cylinder made of FGM with properties modeled by the power law, using the direct method of solution to the Navier equation were obtained by [16]. Steady-state and unsteady temperature and thermal stress distributions in a plate, a hollow circular cylinder, and a hollow sphere made of functionally gradient material have been studied [22,23,24]. They proposed the original analytical method for solution to the one-dimensional heat conductivity problem for heterogeneous FGMs, which was performed with proper displacement of variables, Laplace transform, and the perturbation method. It should be noted that the perturbation method may be employed for the study of all classes of thermoelastic problems for functionally graded materials, even with consideration of the thermal sensitivity of material [11]. In [10], an analytical solution of the heat conduction problem for FGM cylinders subjected to non-uniform heat flux was obtained by using the method of matched asymptotic expansion in the perturbation technique. In [25], the Hankel transform method was used to obtain an analytical solution of the axisymmetric stationary problem of heat conduction for an FGM layer with thermal conductivity dependent on the depth from the heated boundary surface. The same technique has been applied to solve the steady axisymmetric boundary problem of thermoelasticity for non-homogeneous semi-space with thermomechanical properties that depend exponentially on the distance from the heated surface [26]. This approach can be used for modeling layered composites with stepwise gradation of the properties, and also for approximate modeling of the materials with a functional change in properties. In this last case, the functionally graded coatings were replaced by a package of layers, whose material properties were assumed to be constant. This simplification of material property gradation allows one to implement analytical methods for each sublayer, using the known solutions to the thermoelastic problems of friction for isotropic bodies. This approach is known as the multi-layered model of functionally graded materials. It has been shown that the results obtained for the FGM layer, divided into a sufficient number of the sublayers, are in good agreement with the data found using the corresponding exact solutions [26]. The same approach has been used to simulate FGM with sinusoidal and cosinusoidal power and exponential distribution of properties for a cylinder subjected to non-uniform heat flux [27]. Thermoelastic frictional contact of the FGMs with arbitrarily varying properties has been investigated using the multi-layered model by Liu et al. [4,7]. On the basis of the same approach, the non-stationary temperature field in a functional gradient layer with continuous and piecewise change in material properties has been determined by means of the Laplace transform, asymptotic analysis and integration technique [6]. The multi-layered model has been developed for analysis of the two-dimensional sliding frictional contact problem with a functionally graded coating [28]. This model has been used to solve the transient heat conduction and thermal stress problems for the FGM plate taking into consideration temperature-dependent material properties [17].

Most of the above-mentioned studies have considered the problems for a heated FGM layer on homogeneous substrate or cylinder, which can successfully simulate thermoelastic behavior of a brake disc with FGM coating. The temperature mode of a pad-disc tribosystem has been simulated using the thermal problem of friction for a functionally graded coated half-space (a disc) sliding against a homogeneous body (a pad) in [4,7,8,9,27,29]. While modern materials for friction pads in brake systems are usually composites, the proportion of individual components can also be changed along with the distance from the friction surface. Experimental investigations have shown that functional variations in the properties of the pad material significantly improve their braking characteristics [12,30]. In particular, the results have indicated that the wear resistance of a specimen made of a functionally graded material is higher than the wear resistance of its analogue made of a homogeneous material [12]. Therefore, FGMs are real candidates for the role of automotive brake pads [29,31]. In connection with this potential application, we see the need for the development of mathematical models of frictional heating of two element systems of the pad-disc type, both made of functionally graded materials. The development of such models is also associated with the possibility of their use in the study of thermoelastic instability (TEI) due to frictional heating. It is known that the system exhibits TEI in brakes when the sliding speed exceeds a critical value [13]. The emergence of instability is accompanied by the concentration of frictional heating over regions much smaller than the nominal contact region, thus, leading to high localized temperature and contact pressure. The appearance of these so-called hot spots results in various undesirable effects such as material transformations, thermal cracking, and brake fade [20]. The studies concerning the effect of material non-homogeneity on thermal instability in brakes have shown that an FGM disc reduces the susceptibility towards TEI by increasing the critical speed of sliding [8,20,27].

## 2. Statement of the Problem

Consider a heat-conduction problem for two semi-infinite FGM bodies (Figure 1).

It is assumed that:The bodies are related to the coordinate Cartesian system Oxyz, and their initial temperature distribution is homogeneous and equal to the ambient temperature $T_{a}$;At the initial time moment t=0, the bodies are pressed to each other with uniform pressure $p_{0}$ acting parallel to the z axis and simultaneously start sliding with constant relative speed $V_{0}$ in the positive direction of the x axis;Due to friction, on the contact surface z=0 heat is generated, which is absorbed by the elements of friction pair in the form of heat fluxes, causing an increase in their temperature T(z,t) over the initial value $T_{a}$;During frictional heating, the sum $q_{1}+q_{2}$ of intensities of heat fluxes directed from the contact surface z=0 along the normal to the insides of the bodies, is equal to the specific power of friction $q_{0}=fp_{0}V_{0}$, where f is the coefficient of friction. At the same time, the temperatures T on the friction surfaces of both bodies are equal [32,33];Changes in the temperature gradients in the directions x and y are negligible and the gradient in the direction z decreases, along with the distance from the contact surface;Thermal conductivity of materials $K_{l}$ are exponential functions of variable z, and their specific heat $c_{l}$ and density $\rho_{l}$, l=1,2 are constant [34]. Here and further, the lower index l=1 indicates the parameters and quantities relating to the first element, and l=2 to the second element.

On the basis of the above assumptions, the temperature rise $\Theta (z,t)=T(z,t)-T_{a}$ of the friction pair was found as the solution to the following boundary-value problem of heat conduction:(1)$$\frac{\partial }{\partial z}\left[K_{1}(z)\frac{\partial \Theta (z,t)}{\partial z}\right]=c_{1}\rho_{1}\frac{\partial \Theta (z,t)}{\partial t},\;z>0,t>0,$$
(2)$$\frac{\partial }{\partial z}\left[K_{2}(z)\frac{\partial \Theta (z,t)}{\partial z}\right]=c_{2}\rho_{2}\frac{\partial \Theta (z,t)}{\partial t},\;z<0,t>0,$$
(3)$${K_{2}(z)\frac{\partial \Theta (z,t)}{\partial z}|}_{z=0^{-}}-{K_{1}(z)\frac{\partial \Theta (z,t)}{\partial z}|}_{z=0^{+}}=q_{0},\;t>0,$$
(4)$$\Theta (0^{-},t)=\Theta (0^{+},t),\;t>0,$$
(5)Θ(z,t)→0, |z|→∞, t>0,
(6)Θ(z,0)=0,|z|<∞.

Taking into consideration the dependencies:(7)$$K_{l}(z)=K_{l,0}e^{\gamma_{l}\left|z\right|},\;\left|z\right|<\infty ,\;K_{l,0}\equiv K_{l}(0),\;\gamma_{l}\geq 0,l=1,2,$$
the problem Equations (1)–(6) can be written in the form:(8)$$\frac{\partial^{2}\Theta (z,t)}{\partial z^{2}}+\gamma_{1}\frac{\partial \Theta (z,t)}{\partial z}=\frac{e^{-\gamma_{1}z}}{k_{1,0}}\frac{\partial \Theta (z,t)}{\partial t},\;z>0,t>0,$$
(9)$$\frac{\partial^{2}\Theta (z,t)}{\partial z^{2}}-\gamma_{2}\frac{\partial \Theta (z,t)}{\partial z}=\frac{e^{\gamma_{2}z}}{k_{2,0}}\frac{\partial \Theta (z,t)}{\partial t},\;z<0,t>0,$$
(10)$${K_{2,0}\frac{\partial \Theta (z,t)}{\partial z}|}_{z=0^{-}}-{K_{1,0}\frac{\partial \Theta (z,t)}{\partial z}|}_{z=0^{+}}=q_{0},\;t>0,$$
(11)$$\Theta (0^{-},t)=\Theta (0^{+},t),\;t>0$$
(12)Θ(z,t)→0, |z|→∞, t>0
(13)Θ(z,0)=0,|z|<∞
where
(14)$$k_{l,0}=\frac{K_{l,0}}{c_{l}\rho_{l}},\;l=1,2.$$
are the coefficients of thermal diffusivity of materials on the surface of friction z=0.

## 3. Solution to the Problem

The Laplace transform [35]:(15)L[ Θ(z,t);p]≡Θ¯(z,p)=∫0∞Θ(z,t)eptdt, Rep>0.
application to the problem Equations (8)–(13), gives:(16)$$\frac{d^{2}\bar{\Theta }(z,p)}{dz^{2}}+\gamma_{1}\frac{d\bar{\Theta }(z,p)}{dz}-\frac{p}{k_{1,0}}e^{-\gamma_{1}z}\bar{\Theta }(z,p)=0,\;z>0,$$
(17)$$\frac{d^{2}\bar{\Theta }(z,p)}{dz^{2}}-\gamma_{2}\frac{d\bar{\Theta }(z,p)}{dz}-\frac{p}{k_{2,0}}e^{\gamma_{2}z}\bar{\Theta }(z,p)=0,\;z<0,$$
(18)$${K_{2,0}\frac{d\bar{\Theta }(z,p)}{dz}|}_{z=0^{-}}-{K_{1,0}\frac{d\bar{\Theta }(z,p)}{dz}|}_{z=0^{+}}=\frac{q_{0}}{p},\;$$
(19)$$\bar{\Theta }(0^{-},p)=\bar{\Theta }(0^{+},p),$$
(20)$$\bar{\Theta }(z,p)\to 0,\;\left|z\right|\to \infty .$$

Introducing the new variables and dimensionless parameters:(21)$$\xi_{1}=\xi e^{-\gamma_{1}z/2},\;z\geq 0,\;\xi_{2}=\gamma_{\varepsilon }\xi e^{\gamma_{2}z/2},\;z\leq 0,\;\gamma_{\varepsilon }=\gamma^{*}\sqrt{k_{0}^{*}},$$
(22)$$\xi =\frac{2}{\gamma_{1}}\sqrt{\frac{p}{k_{1,0}}},\;\gamma^{*}=\frac{\gamma_{1}}{\gamma_{2}},\;k_{0}^{*}=\frac{k_{1,0}}{k_{2,0}},$$
the following derivatives can be found:(23)$$\frac{d\bar{\Theta }(z,p)}{dz}={\left(-1\right)}^{l}\frac{1}{2}\gamma_{l}\xi_{l}\frac{d\bar{\Theta }(\xi_{l},p)}{d\xi_{l}},\frac{d^{2}\bar{\Theta }(z,p)}{dz^{2}}=\frac{1}{4}\gamma_{l}^{2}\xi_{l}^{2}\frac{d^{2}\bar{\Theta }(\xi_{l},p)}{d\xi_{l}{}^{2}}+\frac{1}{4}\gamma_{l}^{2}\xi_{l}\frac{d\bar{\Theta }(\xi_{l},p)}{d\xi_{l}},l=1,2.$$

Taking into consideration the relations (23), Equations (16) and (17) are brought to the form:(24)$$\frac{d^{2}\bar{\Theta }(\xi_{l},p)}{d\xi_{l}^{2}}-\frac{1}{\xi_{l}}\frac{d\bar{\Theta }(\xi_{l},p)}{d\xi_{l}}-\bar{\Theta }(\xi_{l},p)=0,\;\xi_{l}>0,\;l=1,2$$

The general solution to Equation (24), satisfying the boundary condition (20), has the form:(25)$$\bar{\Theta }(\xi_{l},p)=A_{l}(p)\xi_{l}I_{1}(\xi_{l}),\;l=1,2$$
where $I_{k}(\cdot )$ are the modified Bessel functions of the first kind of the *k*th order, and $A_{l}(p)$ are the unknown functions. After differentiating the solution (25), and taking into consideration the relations (21), (22) and derivative $\left[xI_{1}(x){]}^{\prime }=xI_{0}(x\right)$ [36] (here and further, the symbol ‘ denotes the ordinary derivative), the following is found:(26)$${\frac{d\bar{\Theta }}{dz}|}_{z=0^{+}}=-\frac{\gamma_{1}}{2}A_{1}(p)\xi^{2}I_{0}(\xi ),\;{\frac{d\bar{\Theta }}{dz}|}_{z=0^{-}}=\frac{\gamma_{2}}{2}A_{2}(p){\left(\gamma_{\varepsilon }\xi \right)}^{2}I_{0}(\gamma_{\varepsilon }\xi ).$$

Substituting the derivatives (26) into the boundary conditions (18) and (19), the system of two linear algebraic equations is obtained with respect to the unknown functions $A_{l}(p)$, l=1,2, the solution of which, has the form:(27)$$A_{1}(p)=2\Lambda \frac{I_{1}(\gamma_{\varepsilon }\xi )}{p\gamma_{\varepsilon }\xi^{2}\psi (p)},\;A_{2}(p)=2\Lambda \frac{I_{1}(\xi )}{p{\left(\gamma_{\varepsilon }\xi \right)}^{2}\psi (p)}$$
where
(28)$$\psi (p)=I_{0}(\gamma_{\varepsilon }\xi )I_{1}(\xi )+K_{\varepsilon }I_{0}(\xi )I_{1}(\gamma_{\varepsilon }\xi ),$$
(29)$$K_{\varepsilon }=\frac{K_{0}^{*}}{\sqrt{k_{0}^{*}}},\;K_{0}^{*}=\frac{K_{1,0}}{K_{2,0}},\;\Lambda =\frac{q_{0}}{\gamma_{2}K_{2,0}}.$$

Taking into consideration the forms of variables $\xi_{l},l=1,2$ (21), (22), and functions $A_{l}(p)$, l=1,2 (27)–(29) the solutions (25) are given as:(30)$$\bar{\Theta }(z,p)=2\Lambda e^{-\gamma_{1}z/2}\frac{\varphi_{1}(z,p)}{\Psi (p)},\;z\geq 0,\;\bar{\Theta }(z,p)=2\Lambda e^{\gamma_{2}z/2}\frac{\varphi_{2}(z,p)}{\Psi (p)},\;z\leq 0,$$
(31)$$\begin{array}{c} \varphi_{1}(z,p)=I_{1}(\gamma_{\varepsilon }\xi )I_{1}(\xi e^{-\gamma_{1}z/2}),\;\varphi_{2}(z,p)=I_{1}(\xi )I_{1}(\gamma_{\varepsilon }\xi e^{\gamma_{2}z/2}),\\ \Psi (p)=p\gamma_{\varepsilon }\xi \psi (p). \end{array}$$

Using the Vashchenko–Zakharchenko theorem [37,38], the inverse Laplace transform of the solutions (30) and (31) can be written in the form:(32)$$\Theta (z,t)=2\Lambda e^{-\frac{1}{2}\gamma_{1}z}\left[\frac{\varphi_{1}(z,0)}{\Psi^{\prime }(0)}+\sum_{n=1}^{\infty }\frac{\varphi_{1}(z,p_{n})}{\Psi^{\prime }(p_{n})}e^{-p_{n}t}\right],\;z\geq 0,\;t\geq 0,$$
(33)$$\Theta (z,t)=2\Lambda e^{\frac{1}{2}\gamma_{2}z}\left[\frac{\varphi_{2}(z,0)}{\Psi^{\prime }(0)}+\sum_{n=1}^{\infty }\frac{\varphi_{2}(z,p_{n})}{\Psi^{\prime }(p_{n})}e^{-p_{n}t}\right],\;z\leq 0,\;t\geq 0.$$
where $p_{n}>0$, n=1,2,… are the real roots of the transcendental equation ψ(p)=0 with function ψ(p) (28).

With consideration of the expansions [36]:(34)$$I_{0}(x)=1+\frac{x^{2}}{4}+\frac{x^{4}}{64}+\ldots ,\;I_{1}(x)=\frac{x}{2}+\frac{x^{3}}{16}+\ldots ,$$
from Equation (31), it can be found that:(35)$$\varphi_{l}(z,p)≅\xi^{2}{\tilde{\varphi }}_{l}(z,p),\;l=1,2,\;\Psi (z,p)≅\xi^{2}\tilde{\Psi }(z,p),$$
(36)$${\tilde{\varphi }}_{1}(z,p)=\frac{1}{4}\gamma_{\varepsilon }e^{-\gamma_{1}z/2}\left[1+\frac{1}{8}{\left(\gamma_{\varepsilon }\xi \right)}^{2}\right],\;z\geq 0,\;\varphi_{2}(z,p)=\frac{1}{4}\gamma_{\varepsilon }e^{\gamma_{2}z/2}\left(1+\frac{1}{8}\xi^{2}\right),\;z\leq 0$$
(37)$$\tilde{\Psi }(p)=p\gamma_{\varepsilon }\left[\frac{1}{2}(1+\gamma_{\varepsilon }K_{\varepsilon })+\frac{1}{16}(1+2\gamma_{\varepsilon }K_{\varepsilon }+2\gamma_{\varepsilon }^{2}+K_{\varepsilon }\gamma_{\varepsilon }^{3})\xi^{2}\right],\;\xi^{2}=\frac{4p}{\gamma_{1}^{2}k_{1,0}},\;$$

At p→0, Equations (35)–(37) lead to:(38)$$\frac{\varphi_{1}(0)}{\Psi^{\prime }(0)}=\frac{e^{-\gamma_{1}z/2}}{2(1+\gamma_{\varepsilon }K_{\varepsilon })},\;z\geq 0,\;\frac{\varphi_{2}(0)}{\Psi^{\prime }(0)}=\frac{e^{\gamma_{2}z/2}}{2(1+\gamma_{\varepsilon }K_{\varepsilon })},\;z\leq 0,$$

Using the relation [36]:(39)$$\begin{array}{c} I_{0}(x)=J_{0}(ix),\;I_{1}(x)=-iJ_{1}(ix),\;{J^{\prime }}_{0}(x)=-J_{1}(x),\;{J^{\prime }}_{1}(x)=J_{0}(x)-x^{-1}J_{1}(x),\\ i\equiv \sqrt{-1}, \end{array}$$
where $J_{k}(\cdot )$ are the Bessel functions of the first kind of the *k*th order, and denoted μ≡iξ, the temperature rise (32), (33), with consideration of Equations (22) and (38), can be written in the form:(40)$$\Theta (z,t)=\Lambda e^{-\gamma_{1}z/2}\left[\frac{e^{-\gamma_{1}z/2}}{\left(1+\gamma_{\varepsilon }K_{\varepsilon }\right)}+\frac{4}{\gamma_{\varepsilon }}\sum_{n=1}^{\infty }\frac{{\hat{\varphi }}_{1}(z,\mu_{n})}{\hat{\Psi^{\prime }}(\mu_{n})}e^{-p_{n}t}\right],\;z\geq 0,\;t\geq 0,$$
(41)$$\Theta (z,t)=\Lambda e^{\gamma_{2}z/2}\left[\frac{e^{\gamma_{2}z/2}}{\left(1+\gamma_{\varepsilon }K_{\varepsilon }\right)}+\frac{4}{\gamma_{\varepsilon }}\sum_{n=1}^{\infty }\frac{{\hat{\varphi }}_{2}(z,\mu_{n})}{\hat{\Psi^{\prime }}(\mu_{n})}e^{-p_{n}t}\right],\;z\leq 0,\;t\geq 0,$$
where
(42)$${\hat{\varphi }}_{1}(z,\mu_{n})=J_{1}(\mu_{n})J_{1}(\gamma_{\varepsilon }\mu_{n}e^{-\gamma_{1}z/2}),\;{\hat{\varphi }}_{2}(z,\mu_{n})=J_{1}(\mu_{n})J_{1}(\gamma_{\varepsilon }\mu_{n}e^{\gamma_{2}z/2}),$$
(43)$$\hat{\Psi^{\prime }}(\mu_{n})=\mu_{n}^{2}[(1+\gamma_{\varepsilon }K_{\varepsilon })J_{0}(\mu_{n})J_{0}(\gamma_{\varepsilon }\mu_{n})-(\gamma_{\varepsilon }+K_{\varepsilon })J_{1}(\mu_{n})J_{1}(\gamma_{\varepsilon }\mu_{n})],\;$$
(44)$$p_{n}=0.25k_{1,0}\gamma_{1}^{2}\mu_{n}^{2},\;$$

$\mu_{n}>0,n=1,2,3,\ldots$, are the real roots of the functional equation:(45)$$J_{0}(\gamma_{\varepsilon }\mu )J_{1}(\mu )+K_{\varepsilon }J_{0}(\mu )J_{1}(\gamma_{\varepsilon }\mu )=0.\;$$

On the contact surface z=0 from Equations (40)–(42), we achieve:(46)$$\Theta (t)\equiv \Theta (0,t)=\Lambda \left[\frac{1}{\left(1+\gamma_{\varepsilon }K_{\varepsilon }\right)}+\frac{4}{\gamma_{\varepsilon }}\sum_{n=1}^{\infty }\frac{\hat{\varphi }(\mu_{n})}{\hat{\Psi^{\prime }}(\mu_{n})}e^{-p_{n}t}\right],\;t\geq 0$$
(47)$$\hat{\varphi }(\mu_{n})\equiv {\hat{\varphi }}_{1}(0,\mu_{n})={\hat{\varphi }}_{2}(0,\mu_{n})=J_{1}(\gamma_{\varepsilon }\mu_{n})J_{1}(\mu_{n}),\;$$

Additionally, assuming that the materials of the friction pair are the same ($K_{1,0}=K_{2,0}\equiv K_{0}$, $k_{1,0}=k_{2,0}\equiv k_{0}$, $\gamma_{1}=\gamma_{2}\equiv \gamma$), then, from Equations (21), (22), and (29), it follows that $K_{\varepsilon }=\gamma_{\varepsilon }=1$ and solution (46) and (47) take the form:(48)$$\Theta (t)=2\Lambda \left(\frac{1}{4}-\sum_{n=1}^{\infty }\frac{e^{-p_{n}t}}{\mu_{n}^{2}}\right),\;t\geq 0,$$
where $J_{0}(\mu_{n})\equiv 0$,$p_{n}=0.25k_{0}\gamma^{2}\mu_{n}^{2}$.

Introducing the following dimensionless variables and parameters:(49)$$\zeta =\frac{z}{a},\;\tau =\frac{k_{1,0}t}{a^{2}},\;\gamma_{l}=\frac{\gamma_{l}^{*}}{a},\;l=1,2,\;\Theta_{0}=\frac{q_{0}a}{K_{1,0}},\;\Theta^{*}=\frac{\Theta }{\Theta_{0}}$$
where a is the thickness of the friction pair elements participating in heat absorption. These parameters are closely related to the concept of effective thickness, i.e., the distance from the friction surface where the temperature is equal to 5% of the maximum value [39].

Taking into consideration the notations (49) in Equations (40)–(45), the dimensionless temperature rise can be written as:(50)$$\Theta^{*}(\zeta ,\tau )=\frac{K_{0}^{*}}{\gamma_{2}^{*}}e^{-\gamma_{1}^{*}\zeta /2}\left[\frac{e^{-\gamma_{1}^{*}\zeta /2}}{\left(1+\gamma_{\varepsilon }K_{\varepsilon }\right)}+\frac{4}{\gamma_{\varepsilon }}\sum_{n=1}^{\infty }\frac{\varphi_{1}^{*}(\zeta ,\mu_{n})}{\hat{\Psi^{\prime }}(\mu_{n})}e^{-\lambda_{n}^{2}\tau }\right],\;\zeta \geq 0,\;\tau \geq 0,$$
(51)$$\Theta^{*}(\zeta ,\tau )=\frac{K_{0}^{*}}{\gamma_{2}^{*}}e^{\gamma_{2}^{*}\zeta /2}\left[\frac{e^{\gamma_{2}^{*}\zeta /2}}{\left(1+\gamma_{\varepsilon }K_{\varepsilon }\right)}+\frac{4}{\gamma_{\varepsilon }}\sum_{n=1}^{\infty }\frac{\varphi_{2}^{*}(\zeta ,\mu_{n})}{\hat{\Psi^{\prime }}(\mu_{n})}e^{-\lambda_{n}^{2}\tau }\right],\;\zeta \leq 0,\;\tau \geq 0,$$
where
(52)$$\varphi_{1}^{*}(\zeta ,\mu_{n})=J_{1}(\gamma_{\varepsilon }\mu_{n})J_{1}(\mu_{n}e^{-\gamma_{1}^{*}\zeta /2}),\;\varphi_{2}^{*}(\zeta ,\mu_{n})=J_{1}(\mu_{n})J_{1}(\gamma_{\varepsilon }\mu_{n}e^{\gamma_{2}^{*}\zeta /2}),$$
(53)$$\lambda_{n}=0.5\gamma_{1}^{*}\mu_{n},\;n=1,2,\ldots$$

On the contact surface ζ=0 from Equations (50)–(52), it follows that:(54)$$\Theta^{*}(\tau )\equiv \Theta^{*}(0,\tau )=\frac{K_{0}^{*}}{\gamma_{2}^{*}}\left[\frac{1}{\left(1+\gamma_{\varepsilon }K_{\varepsilon }\right)}+\frac{4}{\gamma_{\varepsilon }}\sum_{n=1}^{\infty }\frac{\varphi^{*}(\mu_{n})}{\hat{\Psi^{\prime }}(\mu_{n})}e^{-\lambda_{n}^{2}\tau }\right],\;\tau \geq 0,$$
where
(55)$$\varphi^{*}(\mu_{n})\equiv \varphi_{1}^{*}(0,\mu_{n})=\varphi_{2}^{*}(0,\mu_{n})=J_{1}(\gamma_{\varepsilon }\mu_{n})J_{1}(\mu_{n}).$$

## 4. An Asymptotic Solution at the Initial Stage of Sliding

At large values of the parameter p of the Laplace integral transform (15), and taking into consideration the asymptotic behavior of the functions [36]:(56)$$I_{k}(x)≅\frac{e^{x}}{\sqrt{2\pi x}},\;k=0,1,$$
from Equations (28) and (31) it can be found that:(57)$$\varphi_{1}(z,p)≅\frac{e^{(1+\gamma_{\varepsilon }-\gamma_{1}z/2)\xi }}{2\pi \xi \sqrt{\gamma_{\varepsilon }}}e^{\gamma_{1}z/4},\;z\geq 0,\;\varphi_{2}(z,p)≅\frac{e^{(1+\gamma_{\varepsilon }+\gamma_{2}z/2)\xi }}{2\pi \xi \sqrt{\gamma_{\varepsilon }}}e^{-\gamma_{2}z/4},\;z\leq 0,$$
(58)$$\Psi (p)≅(1+K_{\varepsilon })\frac{pe^{(1+\gamma_{\varepsilon })\xi }}{2\pi \sqrt{\gamma_{\varepsilon }}}.$$

Substituting Equations (57) and (58) into Equation (30), the transforms of the temperature rise can be presented as:(59)$$\bar{\Theta }(z,p)=\frac{2\Lambda e^{-(1+2\xi )\gamma_{1}z/4}}{(1+K_{\varepsilon })p\xi },\;z\geq 0,\;\bar{\Theta }(z,p)=\frac{2\Lambda e^{(1+2\xi )\gamma_{2}z/4}}{(1+K_{\varepsilon })p\xi },\;z\leq 0,$$

In view of the notation ξ (22), the transformed solutions (59) can be written in the form:(60)$$\bar{\Theta }(z,p)=\frac{\Lambda \gamma_{1}}{\left(1+K_{\varepsilon }\right)}e^{-\gamma_{1}z/4}\frac{e^{-\sqrt{\frac{p}{k_{1,0}}}z}}{p\sqrt{\frac{p}{k_{1,0}}}},\;z\geq 0,\;\bar{\Theta }(z,p)=\frac{\Lambda \gamma_{1}}{\left(1+K_{\varepsilon }\right)}e^{\gamma_{2}z/4}\frac{e^{-\sqrt{\frac{p}{k_{2,0}}}z}}{p\sqrt{\frac{p}{k_{1,0}}}},\;z\leq 0,$$

Using the relation [40]:(61)$$L^{-1}[p^{-3/2}e^{-\alpha \sqrt{p}};t]=2\sqrt{t}\mathrm{ierfc}\left(\frac{\alpha }{2\sqrt{t}}\right),\;\alpha \geq 0,$$
Taking into consideration notations (29) and (49), the dimensionless temperature rise for small values of the Fourier number τ was received as:(62)$$\Theta^{*}(\zeta ,\tau )=\frac{2\gamma_{\varepsilon }K_{\varepsilon }}{\left(1+K_{\varepsilon }\right)}e^{-\gamma_{1}^{*}\zeta /4}\sqrt{\tau }\mathrm{ierfc}\left(\frac{\zeta }{2\sqrt{\tau }}\right),\;\zeta \geq 0,\;0\leq \tau <<1,$$
(63)$$\Theta^{*}(\zeta ,\tau )=\frac{2\gamma_{\varepsilon }K_{\varepsilon }}{\left(1+K_{\varepsilon }\right)}e^{\gamma_{2}^{*}\zeta /4}\sqrt{\tau }\mathrm{ierfc}\left(-\frac{\zeta }{2}\sqrt{\frac{k_{0}^{*}}{\tau }}\right),\;\zeta \leq 0,\;0\leq \tau <<1,$$
where $\mathrm{ierfc}(x)=\pi^{-1/2}e^{-x^{2}}-x\mathrm{erfc}(x),\mathrm{erfc}(x)=1-\mathrm{erf}(x)$, and the erf(x) is the Gaussian error function [36]. On the contact surface ζ=0 from Equations (63) and (62), it can be obtained that:(64)$$\Theta^{*}(\tau )=\frac{2\gamma_{\varepsilon }K_{\varepsilon }}{\left(1+K_{\varepsilon }\right)}\sqrt{\frac{\tau }{\pi }},\;0\leq \tau <<1,$$

Substituting $\gamma_{1}=\gamma_{2}=0$ and $\gamma^{*}=1$ into Equations (62)–(64), the known solutions can be found to determine the dimensionless temperature increase in the homogeneous bodies [41]:(65)$$\Theta^{*}(\zeta ,\tau )=\frac{2K_{0}^{*}}{\left(1+K_{\varepsilon }\right)}\sqrt{\tau }\mathrm{ierfc}\left(\frac{\zeta }{2\sqrt{\tau }}\right),\;\zeta \geq 0,\;0\leq \tau <<1,$$
(66)$$\Theta^{*}(\zeta ,\tau )=\frac{2K_{0}^{*}}{\left(1+K_{\varepsilon }\right)}\sqrt{\tau }\mathrm{ierfc}\left(-\frac{\zeta }{2}\sqrt{\frac{k_{0}^{*}}{\tau }}\right),\;\zeta \leq 0,\;0\leq \tau <<1,$$
(67)$$\Theta^{*}(\tau )=\frac{2K_{0}^{*}}{\left(1+K_{\varepsilon }\right)}\sqrt{\frac{\tau }{\pi }},\;0\leq \tau <<1.$$

## 5. Numerical Analysis

The numerical analysis was performed based on the exact solutions (50)–(55) and the asymptotic Equations (62)–(64). The elements are both made of functionally graded materials in such a way that their friction surfaces z=0 are purely ceramic ZrO_2_ and Al_3_O_2_ and, along the thickness of the elements, they approach the core materials Ti-6Al-4V and TiC, respectively. The thermal properties of component materials are presented in Table 1.

In view of notations (49), Equation (7), describing the change in thermal conductivity of materials with distance from the surface of friction, becomes:(68)$$K_{l}(z)=K_{l,0}K_{l}^{*}(\zeta ),K_{l}^{*}(\zeta )=e^{\gamma_{l}^{*}\left|\zeta \right|},\;\left|\zeta \right|<\infty ,\;l=1,2$$
where the values of the dimensionless gradient parameters can be calculated from the following relation [8]:(69)$$\gamma_{l}^{*}=\ln(K_{l,1}/K_{l,0}),\;K_{l,0}\equiv K_{l}^{*}(0),\;K_{l,1}\equiv K_{l}^{*}(1),\;l=1,2.$$

The formula (69) provides that thermal conductivity changes in a manner suitable for the FGM composition variations from pure ceramic on the friction surface, achieving the pure core material on the effective thickness a (|ζ|=1) inside elements. The effective thicknesses 3.2 mm and 7.7 mm for the first (l=1) and second (l=2) elements, respectively were calculated in accordance with the methodology [39]. Hence, it can be assumed that a=7.7 mm. Then, from Table 1, the following data are taken: $K_{1,0}=2.09\;{\mathrm{Wm}}^{-1}K^{-1}$, $K_{1,1}=7.5\;{\mathrm{Wm}}^{-1}K^{-1}$ for the FGM ZrO_2_–Ti-6Al-4V (*l* = 1) and $K_{2,0}=1.5\;{\mathrm{Wm}}^{-1}K^{-1}$, $K_{2,1}=33.9\;{\mathrm{Wm}}^{-1}K^{-1}$ for the FGM Al_3_O_2_–TiC (*l* = 2). Substituting these coefficients into the formula (69) we obtain the dimensionless gradient parameters values $\gamma_{1}^{*}=1.28$, $\gamma_{2}^{*}=3.12$. Distribution of the thermal conductivity along the distance from the friction surface, for considered tribosystem is presented in the Figure 2. The positive roots of the nonlinear functional Equation (45) were found by means of the bisection method [43]. It was necessary to take at least 70 roots of Equation (45) in order to perform calculations according to Equations (50)–(55) with a relative accuracy of $10^{-3}$.

Variations of the dimensionless temperature rise $\Theta^{*}(\zeta ,\tau )$ (50)–(55) in the friction elements ZrO_2_–Ti-6Al-4V (l=1) and Al_3_O_2_–TiC (l=2) during the sliding, are shown by the continuous curves in Figure 3, while the dashed lines in this figure illustrate the corresponding results obtained from the solutions (65)–(67) for the friction pair elements made of homogeneous materials ZrO_2_ (l=1) and Al_3_O_2_ (l=2). At a certain distance ζ, the temperature monotonically increases over time (Fourier number τ). The highest temperature is achieved on the contact surface ζ=0. It can be seen that the elements of tribocouple made of homogeneous materials are heated more intensively during the sliding than the FGMs. Differences between the compared results increase over the time of heating. Taking into consideration notations (49), it can be established that, the maximum temperature rises are $\Theta_{\max}={604\;}^{\circ }C$ and $\Theta_{\max}={765\;}^{\circ }C$ achieved at the end of the process, for the friction pairs made of functionally graded and homogeneous materials, respectively.

Distribution of dimensionless maximum temperature $\Theta_{\max}^{*}$, achieved at the end of the process, along the distance from the contact surface is presented in Figure 4. With the distance from the contact surface in the element l=1, the difference between continuous and dashed lines decreases. Unlike in the element l=2, where this difference remains almost unchanged along the thickness, and even slightly increases (to the $238^{\circ }C$ at distance |z|=1.85 mm). A much higher temperature level is reached in the homogeneous element l=2 made of ceramic (Al_3_O_2_) as compared with the temperature achieved in the FGM element Al_3_O_2_–TiC, which is caused by application of the core material (TiC) with high thermal conductivity and diffusivity.

The time profiles of dimensionless temperature rise $\Theta_{}^{*}$ on the contact surface ζ=0 for different values of the parameter $\gamma_{l}^{*}$, l=1,2, are demonstrated in Figure 5. At a certain moment of time, the temperature of the friction surface increases with a decrease in the material gradient parameter (the continuous curves), approaching the temperature values obtained for a friction pair made of homogeneous materials (the dashed curves). The differences between the individual curves obtained for FGMs and homogeneous materials grow with the time of sliding.

The influence of dimensionless gradients of materials $\gamma_{l}^{*}$, l=1,2 (69) on the dimensionless maximum temperature $\Theta_{\max}^{*}$ of the contact surface is shown in Figure 6 (the continuous curves). The dashed lines in this figure present the corresponding data calculated on the basis of the solution found for the homogeneous materials. The highest values $\Theta_{\max}^{*}$ are achieved for the elements of the friction pair made of the homogeneous ceramic materials. Increasing the gradient parameters of the material reduces the maximum temperature of the friction system. The highest reduction of $\Theta_{\max}^{*}$ takes place while increasing the parameter $\gamma_{1}^{*}$ in the element made of ZrO_2_–Ti-6Al-4V, while the parameter $\gamma_{2}^{*}$ value of the element Al_3_O_2_–TiC remains constant (Figure 6a).

Distributions of dimensionless temperature $\Theta^{*}$ at the end of the sliding, along the distance 0≤|ζ|≤1 from the friction surface is demonstrated in Figure 7. The highest distance value |ζ|=1 corresponds to the previously established maximum effective thickness of heating a=7 mm. As agreed, so far, on the contact surface ζ=0, the temperature of the friction pair made of FGMs is lower than that in the case of the tribocouple with homogeneous materials. Increasing the distance from the contact surface reduces the temperature in both cases, the system made of FGMs (continuous curves) and the friction pair made of ceramic homogeneous materials (dashed curves).

The temperature in the first element (l=1) decreases faster than that in the case of the homogeneous material ZrO_2_, while in the second element (l=2), the temperature of the homogeneous material Al_3_O_2_ remains higher than the temperature of the element made of FGM Al_3_O_2_–TiC, throughout the whole effective thickness. At a certain distance from the friction surface, increasing the material gradient parameters (enhancement of the volume fraction of the core material in the composite structure) causes a drop of the temperature in both FGMs used.

## 6. Conclusions

According to the obtained solutions, the numerical analysis of the temperature mode was performed for friction pair elements made of functionally graded materials, under uniform sliding. The friction surfaces of these elements are ceramic materials, i.e., zirconium dioxide ZrO_2_ and aluminum oxide Al_3_O_2_. The volume fraction of ceramics in the materials decreases with the depth, in favor of the core materials.

The composites are the high-class titanium-aluminum-vanadium alloy Ti-6Al-4V and titanium carbide TiC, with higher thermal conductivities than ceramics. On the basis of the results of the calculations, the influences of the values of the friction material gradient parameters on the time-space temperature distributions in the tribological system were investigated. The obtained data show that the use of selected composites with a continuous (exponential) change of thermal conductivity, improves the friction conditions, causing a significant decrease in the temperature level reached on the friction surface, especially the maximum value at the end of the sliding.

Despite its purely theoretical importance, the determined analytical solution also has practical significance. On the basis of this closed-formed expression, it is possible to quickly estimate the temperature mode of a friction system made of FGMs with an exponential gradient under uniform sliding. Furthermore, the exact solutions play the role of a template for testing the approximate numerical methods. It should be noted that the solution of formulated thermal problem of friction was obtained assuming an exponential change in thermal conductivity. Thus, the developed model is oriented only to the FGM class with just such a gradient. In this sense, it is a natural limitation of the solution. Other application limitations of this model (unidirectional heating process, ideal thermal contact of bodies, etc.) are presented in the assumptions.

In the next report, we plan to present the results concerning a study of the impact of functional gradient structure of friction materials on the temperature in a disc brake system during braking.

## Figures and Tables

**Figure 1 materials-14-04285-f001:**
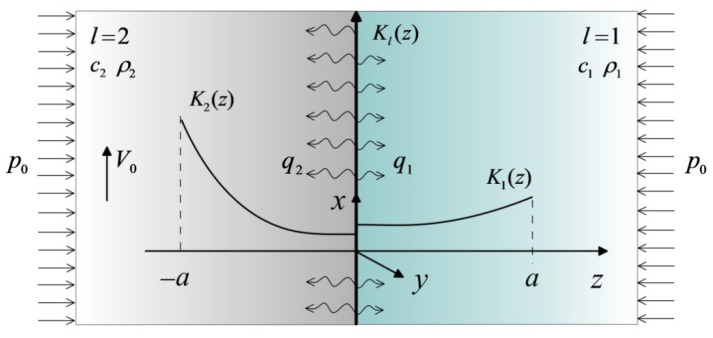
Scheme of the problem.

**Figure 2 materials-14-04285-f002:**
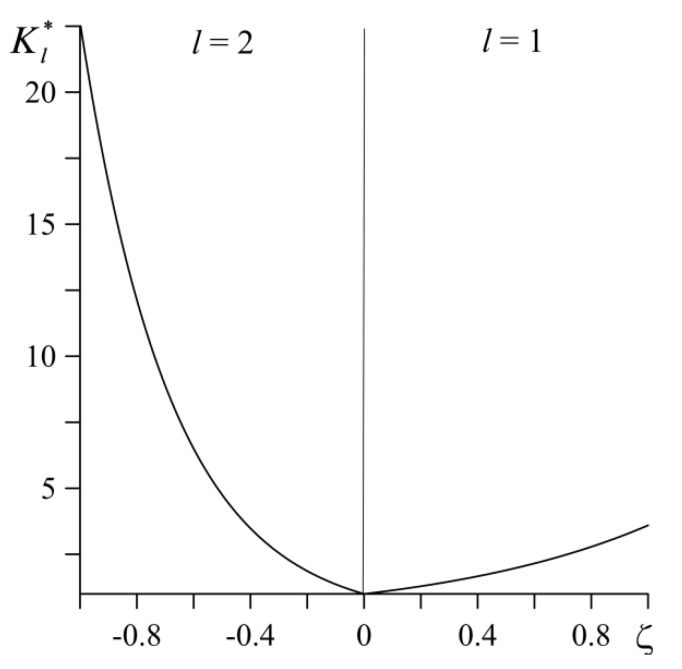
Distributions of the dimensionless thermal conductivities $K_{l}^{*}$, of FGM ZrO_2_–Ti-6Al-4V (*l* = 1) and Al_3_O_2_–TiC (*l* = 2) along the dimensionless distance ζ from the friction surface.

**Figure 3 materials-14-04285-f003:**
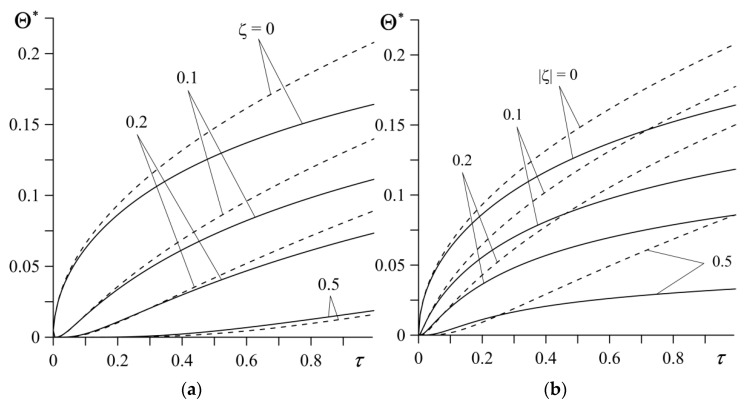
Evolution of the dimensionless temperature $\Theta^{*}(\zeta ,\tau )$ during sliding at different distances from the friction surface. Continuous curves represent FGMs: (**a**) ZrO_2_–Ti-6Al-4V; (**b**) Al_3_O_2_–TiC and the dashed curves represent homogeneous materials: (**a**) ZrO_2_; (**b**) Al_3_O_2_.

**Figure 4 materials-14-04285-f004:**
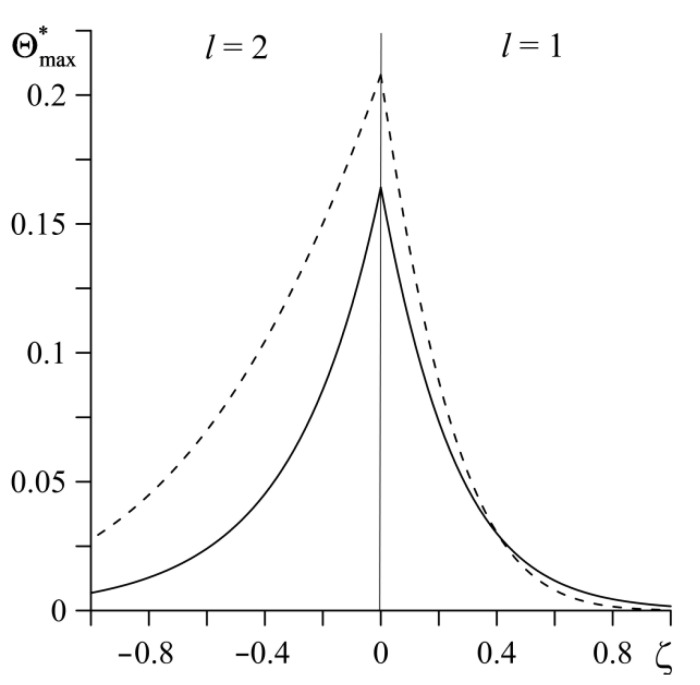
Distribution of the dimensionless maximum temperature rise $\Theta_{\max}^{*}$ reached at the end of friction process, along the distance ζ from the friction surface. Continuous curves represent FGMs ZrO_2_–Ti-6Al-4V (*l* = 1) and Al_3_O_2_–TiC (*l* = 2); the dashed curves represent homogeneous materials ZrO_2_ (*l* = 1) and Al_3_O_2_ (*l* = 2).

**Figure 5 materials-14-04285-f005:**
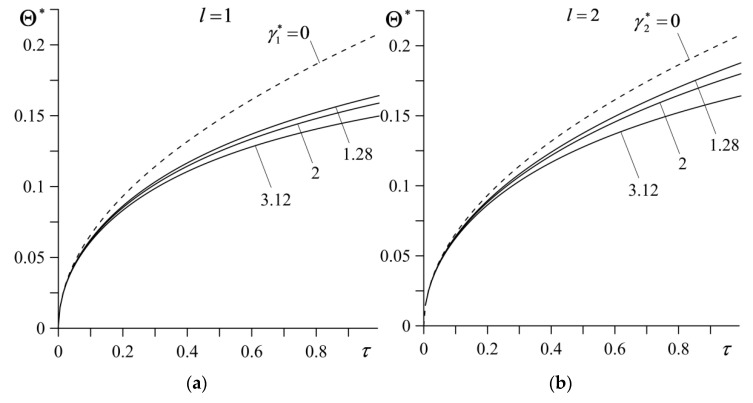
Evolutions of dimensionless temperature rise $\Theta_{}^{*}$ on the contact surface of the friction pair for various values of parameter: (**a**) $\gamma_{1}^{*}$ for $\gamma_{2}^{*}=3.12$; (**b**) $\gamma_{2}^{*}$ for $\gamma_{1}^{*}=1.28$. Continuous curves represent FGMs ZrO_2_–Ti-6Al-4V (*l* = 1) and Al_3_O_2_–TiC (*l* = 2), the dashed curves represent homogeneous materials ZrO_2_ (*l* = 1) and Al_3_O_2_ (*l* = 2).

**Figure 6 materials-14-04285-f006:**
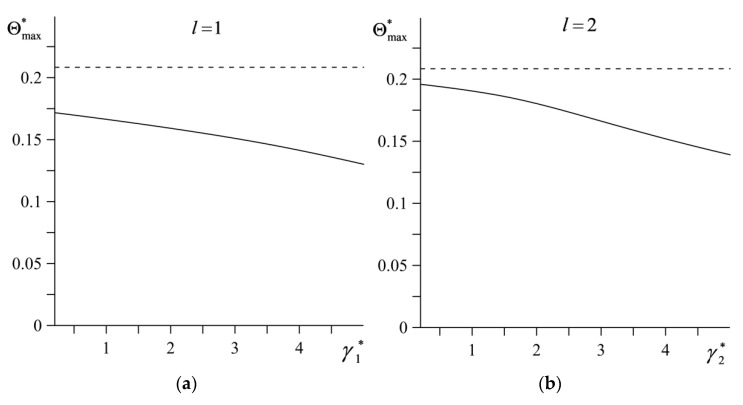
Dependencies of the maximum dimensionless temperature rise $\Theta_{\max}^{*}$ on the dimensionless gradient of material: (**a**) $\gamma_{1}^{*}$ for $\gamma_{2}^{*}=3.12$; (**b**) $\gamma_{2}^{*}$ for $\gamma_{1}^{*}=1.28$. Continuous curves represent FGMs ZrO_2_–Ti-6Al-4V (*l* = 1) and Al_3_O_2_–TiC (*l* = 2), the dashed curves represent homogeneous materials ZrO_2_ (*l* = 1) and Al_3_O_2_ (*l* = 2).

**Figure 7 materials-14-04285-f007:**
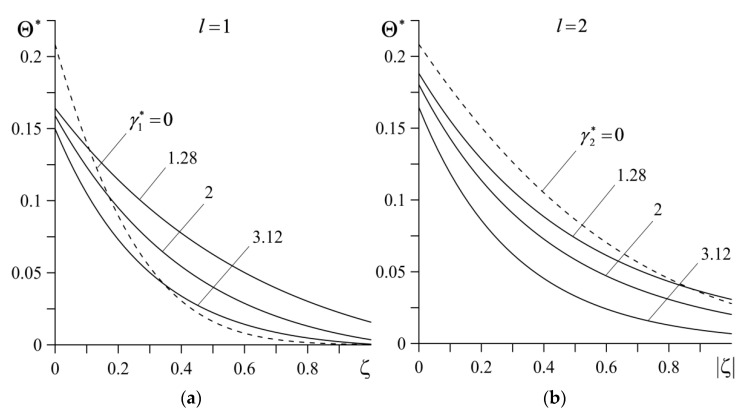
Dependencies of dimensionless temperature rise $\Theta^{*}$ at the end of heating, on the dimensionless distance |ζ| from the contact surface for different values of parameters: (**a**) $\gamma_{1}^{*}$ for $\gamma_{2}^{*}=3.12$; (**b**) $\gamma_{2}^{*}$ for $\gamma_{1}^{*}=1.28$. Continuous curves represent FGMs ZrO_2_–Ti-6Al-4V (*l* = 1) and Al_3_O_2_–TiC (*l* = 2) and the dashed curves represent homogeneous materials ZrO_2_ (*l* = 1) and Al_3_O_2_ (*l* = 2).

**Table 1 materials-14-04285-t001:** Thermal properties of the FGMs components [17,42].

Element Number.	Material	$\mathrm{Thermal}\;\mathrm{Conductivity}\;K[Wm^{-1}K^{-1}]$	$\mathrm{Thermal}\;\mathrm{Diffusivity}\;k\;\times \;10^{6}[m^{2}s^{-1}]$
l=1	ZrO_2_	2.09	0.86
	Ti-6Al-4V	7.5	3.16
l=2	Al_3_O_2_	1.5	4.98
	TiC	33.9	9.59

## Data Availability

No new data were created or analyzed in this study. Data sharing is not applicable to this article.

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
