# Peer review of "The Effect of Functionally Graded Materials on Temperature during Frictional Heating: Under Uniform Sliding"

_materials, 2021, doi:10.3390/ma14154285_

Round 1

Reviewer 1 Report

Line 180: Please think about the following. If OX axis is oriented along the contact surfaces, one fixed and other moving with the speed V0, then the OX axis will be always at the temperature  of the surfaces in contact. These  temperatures rises continuously until the system stabilizes in a stationary temperature regime. Didn't you hurry to say that the temperature gradient on the OX axis in negligible???

Table 1: The fact that you offer conductivity and diffusivity of the materials that make up the FGM is necessary, but just as necessary was the presentation of the variation of these thermal properties for a concrete case of FGM. How does the composition gradient vary and how does this variation influence the  thermal properties in depth?

Line 262: The theoretical relation (68) must be supported with numerical values of various FGMs with different  compositions gradient. Otherwise, the proposed method  remains strictly  theoretical and is not related to concrete practical situations. i.e. is neglected exactly the material announced in the title, the one  arouses interest in  the proposed calculations.

Line 270: Please exemplify how to switch from the numeric values from tab.1  to the numeric values from line 271? What type of FGM  was considered, with what composition gradient? The composition gradient determines a corresponding gradient of the values ϒ*1 and ϒ*2. Here the proposed process cases confusions because not link the depth variation of the thermal properties with the depth variation of the chemical composition.

Line 285: You must indicate the correspondence between calculated and measured temperature values. How was their measurement made?

Fig.3: The values plotted for ϒ*1 and ϒ*2 are valid for the given FGM composition. Show this composition. Or are they just theoretical and not related to a particular material? Where is the dependence of ϒ*1 and ϒ*2 even they are theoretical, i=on the depth of the layer in which the temperature variation was analyzed?

Bibliography: Out of 40 titles only 9 are from the last 5 years. Please update references. In positions: 2, 12, 32, 35, 36, 38, 39, 40, the year of publication must be "bold" s so for the other titles.

General observation:  Try to be more concrete in linking the theory to the variation of the properties of a depth FGM. Otherwise you offer a mathematical model that goes with ny type of material. If you are doing this, please know that it is not visible to a reader.

Reviewer 2 Report

The authors consider the unidimensional (in space) problem and after  the rather cumbersome process get analytical solutions (50-55) and (62-64). Only exponential change of thermal conductivity is considered.  So, it is not entirely clear what limitations the proposed analytical technique has.  In Introduction, some other (power-law, sinusoidal, and piecewise constant) distributions are mentioned.

Due to combersome expressions containing special functions (expressed via infinite series), it looks like the analytical solutions obtained are not better than numerical ones from the computation point of view. It is not difficult to solve problems (1-6) and (7-13) numerically because we do not need to build a 2D or 3D spatial FEM grid.

So, it would be interesting to know more about advantages of the analytical solution proposed by the authors.

Minor errors also should be corrected:

  • In Eq. (2) one must change "z>0" for "z<0";
  • superscript "2" is omitted in the second partial derivation with respect to variable z in Eq. (9);
  • subscript "2" instead of "1" must be in the second relation in Eq. (57);
  • what does the absolute value of dzeta mean in Fig. 2b?
  • must be "takes place" in row 322.

Round 2

Reviewer 1 Report

Now the paper has enough quality to deserve to be published